# Metabolomics of developmental changes in *Triatoma sanguisuga* gut microbiota

**Evan Teal** [ID]**, Claudia Herrera, Eric Dumonteil** [ID]*

Department of Tropical Medicine, School of Public Health and Tropical Medicine, and Vector-Borne and Infectious Disease Research Center, Tulane University, New Orleans, Louisiana, United States of America

* edumonte@tulane.edu

## Abstract

*Triatoma sanguisuga* is one of the major vectors of *Trypanosoma cruzi* in the southeastern US, where it sustains a robust zoonotic parasite transmission cycle and occasional human infections. A better understanding of triatomine development may allow for alternative approaches to insecticide-based vector control. Indeed, the role of the gut microbiota and bacterial endosymbionts in triatomine development and in their vectorial capacity is emerging. We investigated here the differences in microbiota among nymph and adult *T. sanguisuga*, to shed light on the metabolomic interactions occurring during development. Microbiota composition was assessed by 16s gene amplification and deep sequencing from field-caught adult bugs and their laboratory-raised progeny. Significant differences in microbiota bacterial diversity and composition were observed between nymphs and adults. Laboratory-raised nymphs showed a higher taxonomic diversity, and at least seven families predominated. On the other hand, field-caught adults had a lower bacterial diversity and four families comprised most of the microbiota. These differences in compositions were associated with differences in predicted metabolism, with laboratory-raised nymphs microbiota metabolizing a limited diversity of carbon sources, with potential for resource competition between bacterial families, and the production of lactic acid as a predominant fermentation product. On the other hand, field-caught adult microbiota was predicted to metabolize a broader diversity of carbon sources, with complementarity rather than competition among taxa, and produced a diverse range of products in a more balanced manner. The restricted functionality of laboratory-raised nymph microbiota may be associated with their poor development in captivity, and further understanding of the metabolic interactions at play may lead to alternative vector control strategies targeting triatomine microbiota.

**Data Availability Statement:** Raw sequence data are available in the NCBI SRA database under BioProject PRJNA892946, BioSamples SAMN31403430-.SAMN31403445.

## Introduction

Chagas disease is a parasitic illness which afflicts 6–8 million people in Latin America alone [1], and an estimated 288,000 people in the United States [2]. It is caused by *Trypanosoma cruzi* parasites, which are transmitted primarily by triatomine insect vectors, although additional transmission routes contribute to disease epidemiology. At least 11 triatomine species

**Funding:** The author(s) received no specific funding for this work.

**Competing interests:** The authors have declared that no competing interests exist.

can transmit *T. cruzi* in the United States, with *Triatoma sanguisuga* the most relevant species in the southeastern part of the country [3]. It is responsible for an active zoonotic parasite transmission cycle involving multiple sylvatic and synanthropic mammalian hosts, with frequent spill-over in domestic animals including pet dogs and cats [4–7], and occasional cases of autochthonous human cases, as seen in Mississippi [8], Tennessee [9] and Louisiana [10].

Vector control conventionally relies on residual insecticide spraying, which shows limitation in the case of intrusive sylvatic vectors [11], and alternative approaches are critically needed. In that sense, triatomine gut microbiota and the key contribution of specific bacterial endosymbionts to bug development and to its vectorial capacity are emerging as attractive targets for novels strategies. For example, the successful development of *Rhodnius prolixus*, one of the most epidemiologically relevant vector species in large parts of South America [12, 13], is thought to be critically dependent on the provision of B-complex vitamins by its endosymbiont *Rhodococcus rhodnii*, which is acquired during bug development through coprophagy [14]. Thus, vitamin B deficiency in laboratory colonies leads to growth retardation and nymphs fail to reach the adult stage in the absence of *R. rhodnii*, although a more recent study suggests that additional bacteria in the gut microbiota may also contribute to vitamin B production [15]. Furthermore, other studies suggest that the microbiota is likely providing multiple metabolites in addition to vitamin B [16].

In spite of these studies, our understanding of the contribution of the microbiota to triatomine development remains limited. A general trend in field-caught bugs is that nymphs have a highly diverse microbiota, which often differs from that of adult bugs, suggesting different metabolic functions and interactions. This is the case for *Triatoma protracta* [17], *Triatoma reticularia* [17], *Triatoma rubida* [17], *Triatoma dimidiata* [18, 19], and *Rhodnius prolixus* [20]. On the other hand, in *Triatoma sordida* the adult microbiota is more diverse compared to nymphs, with unique bacteria being incorporated from the 4th instar stage onwards [21]. Furthermore, it is still unclear how such changes in microbiota composition may translate into differences in metabolic profiles and interactions among bacteria and the bugs. In *T. sanguisuga*, too limited data is available to assess changes in microbiota during development [17], but this species is notoriously difficult to raise in captivity as nymphs mostly stop molting in the third instar stage and die [22, 23]. This has been associated with the requirement of maintaining nymphs together with field-collected adults, under the hypothesis that this allowed them to acquire necessary flora through coprophagy [23].

The gut microbiota also likely interacts with *T. cruzi* parasites as these reside in the triatomine gut, and the microbiota may modulate vectorial capacity as bacteria and parasite compete for metabolites and may inhibit one another. For example *Serratia marcescens*, another member of *R. prolixus* microbiota, has a trypanolytic activity that can antagonize *T. cruzi* growth [24, 25]. In fact, *T. cruzi* infection leads to changes in microbiota composition in several triatomine species, suggesting tight interactions among bacteria and the parasite [26–29]. These interactions may also be highly parasite strain-specific, as some differences in microbiota composition were detected according to *T. cruzi* parasite DTU present in *T. sanguisuga* [4] and *T. dimidiata* [29]. The introduction of transgenic organisms into the microbiota of triatomine vectors through various strategies has been proposed before as a potential approach to promote the production of trypanolytic factors by endosymbionts to reduce *T. cruzi* establishment in vectors, hence reducing vector capacity [30, 31], and a better understanding on microbiota functions may lead to effective targets.

Because *T. sanguisuga* may be an interesting model to better evaluate the role of the microbiota in triatomine development, we sought to assess potential differences in microbiota between nymphs and adults, to test the hypothesis that laboratory-raised nymphs lack key bacteria compared to field-collected adults [23]. We further predict key functional metabolic

properties of the identified microbiota as a first step towards evaluating microbiota metabolic functions in *T. sanguisuga*.

## Materials and methods

### Triatomines and DNA extraction

Insects were collected across Louisiana, USA, through community participation, in the years 2019 and 2021, and brought to an biosafety level 2 facility for diagnosis of *T. cruzi* in the insect feces via PCR. Uninfected adult females were kept individually in Nalgene containers with mesh lined tops and filter paper folds to lay eggs. Bugs were fed every 2–3 weeks via bell feeder using sheep blood. Eggs and hatched nymphs were kept in the same containers with the adults. All nymphs failed to molt past the third instar stage and died. All bugs were collected and stored at -20ºC until used. A total of 23 insects were used: 18 nymphs (8 first instar, 6 second instar, and 4 third instar) and 5 female adults. For adults, DNA was extracted from the distal section of insect abdomens using a sterile disposable scalpel cleaned with dilute bleach, whereas for nymph the entire insect was used. DNA was extracted using the Qiagen DNEasy blood and tissue extraction kit following manufacturer's instructions.

### 16s RNA gene amplification and sequencing

The full-length 16s RNA gene (about 1500 bp) was amplified using Oxford Nanopore™ 16s Barcoding kit (SQKRAB204) following the manufacturer's instructions except that two rounds of PCR amplification were performed for library preparation. DNA libraries were sequenced on a Nanopore MinIon platform. Raw sequence data are available in the NCBI SRA database under BioProject PRJNA892946, BioSamples SAMN31403430-.SAMN31403445.

### Sequence and data analysis

Fastq DNA sequences were filtered for quality and length and analyzed with a Bayesian classifier from the Ribosomal Database Project [32] as implemented in Geneious Prime. We used a threshold of ≥97% sequence identity for taxonomic identification of bacterial taxa at the family level. Microbiota composition was then analyzed using MicrobiomeAnalyst [33, 34]. Data were first rarefied and normalized using the Total Sum Scaling (TSS) method, to account for variability in sequencing depth. Chao1 and Shannon alpha diversity indices were calculated and compared between groups using t-tests. For beta diversity, Non-Metric Dimensional Scaling (NMDS) was used based on Bray-Curtis distances and statistical significance of differences was assessed by Permutational ANOVA (PERMANOVA). Associations of individual bacterial families with triatomine developmental stage were assessed by correlation analysis. A co-occurrence network of bacterial families based on SparCC correlations was also constructed to identify bacteria which presence/absence are strongly correlated within a specific developmental stage of *T. sanguisuga*. FDR corrections were used to account for multiple testing in establishing the statistical significance of the associations.

For metabolic analysis, the AGORA database from the Virtual Metabolic Human was used, which provides metabolic reconstruction of over 800 bacterial species from microbiomes [35]. Carbon sources and fermentation products associated with the most abundant bacterial families identified in *T. sanguisuga* were extracted and weighted based of the proportion of species within each family able to use/produce each metabolite. Seven bacterial families were included for nymphs, and four for adults, which accounted for over 85% of the microbiota for each group. The relative level of each metabolite used/produced by each bacterial family was

calculated based on their relative abundance in the microbiota, to establish the predicted integrated metabolic profile of the bacterial community.

## Results

### Microbiota composition of nymph and adult *T. sanguisuga*

A total of 23 bugs were included in the study, but data from seven first instars were discarded from the analysis due to poor DNA extraction, 16s amplification or sequencing. The remaining 16 samples yielded an average of 80,947 reads per bug. After filtering and normalization, rarefaction curves indicated that this sequencing amount was sufficient to recover most of the bacterial diversity of these samples (S1 Fig). An initial comparison of second and third instar microbiota composition indicated that their alpha and beta diversity were not significantly different (S2 Fig), and these were combined into a single nymph group for further comparison with adult microbiota.

Comparison of alpha diversity between *T. sanguisuga* laboratory-raised nymph and field-caught adult microbiota indicated that nymphs harbored a greater diversity of bacterial families than adults, as indicated by Shannon and Chao1 indices, although the later did not reach statistical significance (Fig 1A and 1B, t = 2.88, $P$ = 0.017 and t = 0.29, $P$ = 0.78, respectively). Indeed, laboratory-raised nymph's microbiota included up to 14 bacterial families, while only 10 were detected in field-caught adults (Fig 1C). Seven taxa comprised over 85% of the laboratory-raised nymph microbiota, including Xanthomonadaceae, Moraxellaceae, Clostridales incertae sedis, Propionibacteriaceae, Staphylococcaceae, Burkholderiales incertae sedis and Aerococcaceae, while the field-caught adult microbiota included four predominant taxa: Enterobacteriaceae, Porphyromonadaceae, Staphylococcaceae and Peptoniphilaceae. Analysis of beta diversity confirmed that there was a significant difference in microbiota composition between nymph and adult bugs (Fig 2, F = 4.2; $R^2$ = 0.23; $P$ = 0.002).

Four families rather abundantly present in laboratory-raised nymphs were absent in field-caught adults (Burkholderiales incertae sedis, Carnobacteriaceae, Comamonadaceae and Moraxellaceae), and an additional six families present in both nymphs and adults were strongly decreased in adults (Aerococcaceae, Burkholderiaceae, Clostridales incertae sedis, Propionibacteriaceae, Streptococcaceae and Xanthomonadaceae). On the other hand, Enterobacteriaceae, Porphyromonadaceae, Staphylococcaceae, which were minor component of the laboratory-raised nymph microbiota, were strongly increased and predominated in the microbiota of field-caught adults.

### Associations among bacterial families and developmental stage

To further assess which bacteria may associate with specific developmental stage, we performed correlation analysis. Peptoniphilaceae, Porphyromonadaceae and to a lesser extent Enterobacteriaceae were significantly associated with the field-caught adult microbiota, while Carnobacteriaceae, Clostridales incertae sedis, and Streptococcaceae were significantly associated with laboratory-raised nymph microbiota (Fig 3). Analysis of co-occurrence networks also indicated that several bacterial families were occurring simultaneously in a microbiota, suggesting functional interactions. For example, Peptoniphilaceae and Porphyromonadaceae were frequently found together in field-caught adults, while Propionibacteraceae and Clostridales insertae sedis co-occurred in laboratory-raised nymphs (Fig 4). Similarly, Moraxellaceae, Burkholderiales incertae sedis and Comamonadaceae co-occured in nymphs and may antagonize Staphylococcaceae which was more abundant in adults.

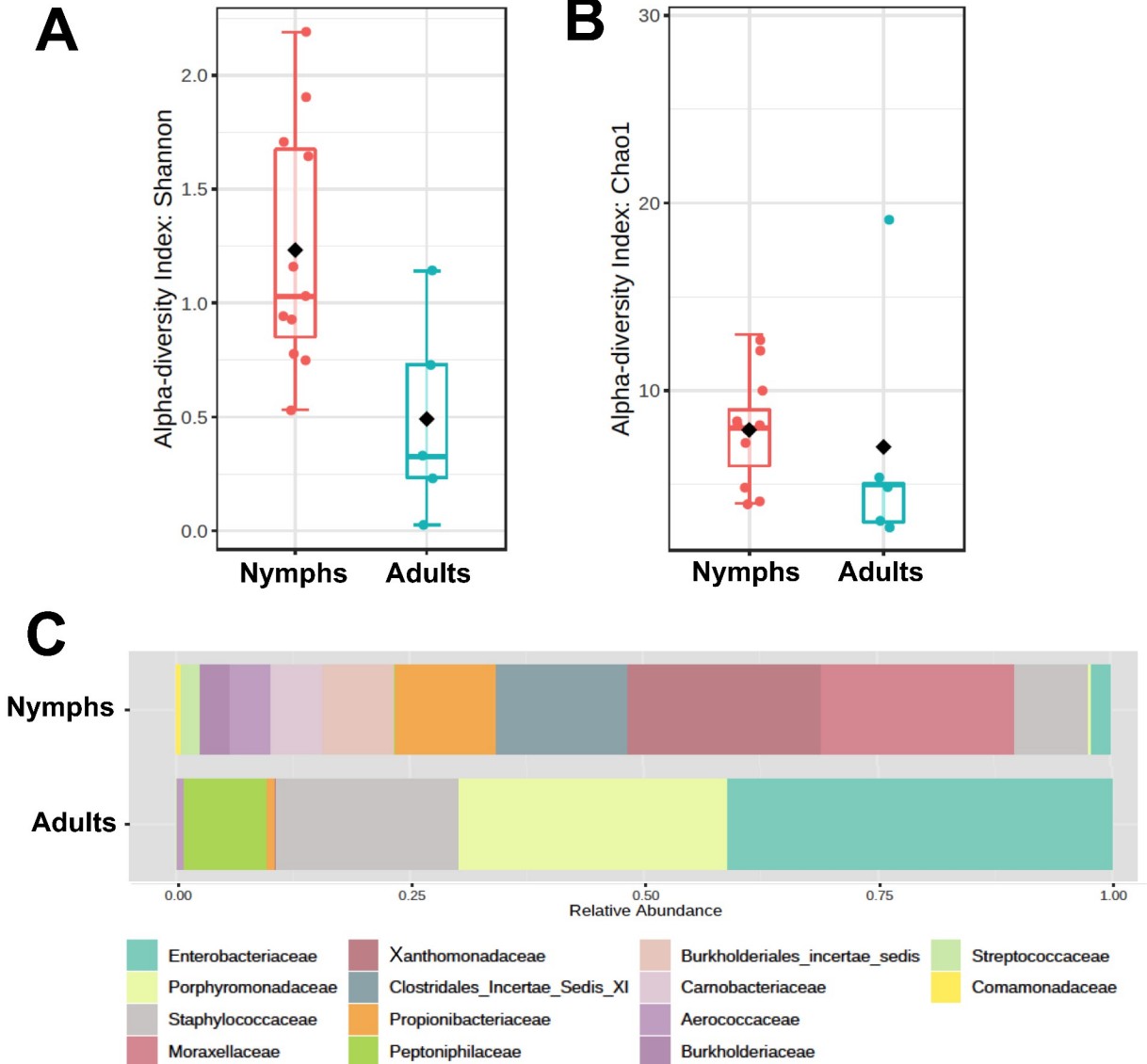

**Fig 1. Diversity of *T. sanguisuga* microbiota during development.** Box plots and individual values of alpha diversity expressed as Shannon (**A**) and Chao1 indices (**B**) were compared for laboratory-raised nymphs and field-caught adults. Nymphs presented a significantly higher Shannon index compared to adults (t = 2.88, *P* = 0.017), but the difference in Chao1 index did not reach statistical significance (t = 0.29, *P* = 0.78). (**C**) Bacterial composition of laboratory-raised nymph and field-caught adult microbiota. Taxonomic groups are color-coded as indicated.

## Metabolic function of *T. sanguisuga* microbiota

The contribution of the predominant bacterial families to the integrated microbiota metabolic functions was then predicted, based on the AGORA database [35]. We included metabolic functions from seven families in nymphs and four in adults, which represented over 85% of their respective microbiota. Bacteria from the laboratory-raised nymph microbiota used a limited variety of carbon sources, in which glucose largely predominated, and the use of the same carbon sources by multiple bacterial families suggested potential competition for these resources (Fig 5A). The fermentation products of the laboratory-raised nymph microbiota included mostly lactic and acetic acid, and a limited variety of additional compounds in lesser

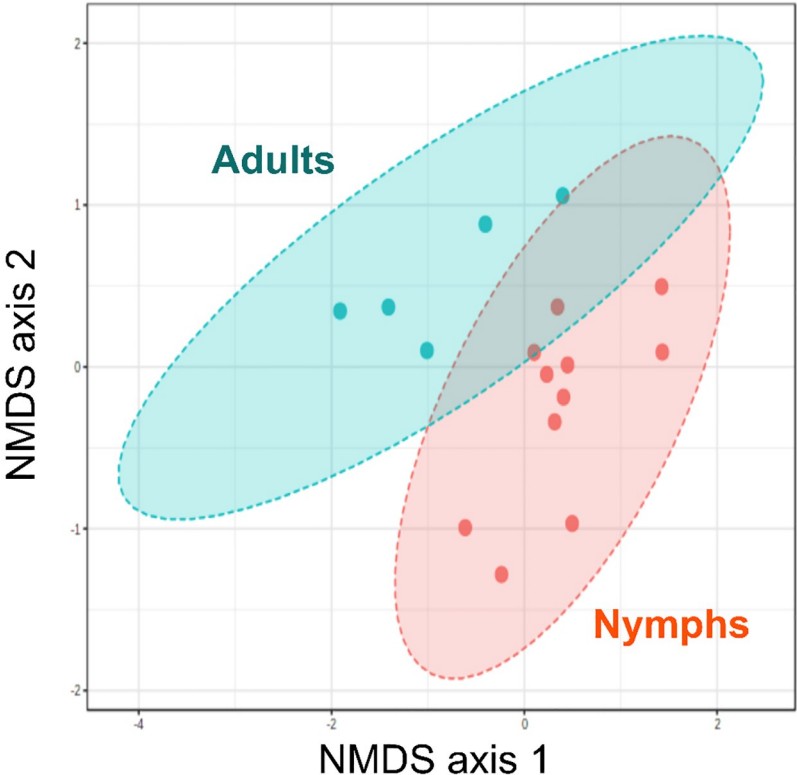

**Fig 2. Beta diversity of *T. sanguisuga* microbiota.** The beta diversity was compared between laboratory-raised nymphs and field-caught adults through NMDS analysis, which showed a statistically significant difference in taxonomic composition of the microbiota (PERMANOVA, F = 4.2; $R^2$ = 0.23; $P$ = 0.002). Points indicate individual bugs, and 95% elipses are shown for each group.

amounts (Fig 5B). On the other hand, the field-caught adult microbiota, in spite of comprising a limited diversity of bacteria, appeared able to process a broader diversity of carbon sources in a more balanced manner, even though glucose was also an important source (Fig 5C). Also, while the adult microbiota fermentation products included lactic and acetic acids, it was not as biased as that of laboratory-raised nymphs, and a larger diversity of compounds were also produced in more comparable relative amounts (Fig 5D). For example, indole, (2R,3R)-2,3-buta-nediol and phenylacetic acid were only produced by the adult microbiota. Together, these data point to important differences in metabolic functions of the microbiota between laboratory-raised nymphs and field-caught adults, with the microbiota from adults presenting potential functional gains in terms of metabolic properties.

## Discussion

Triatomine gut microbiota is thought to play a key role in bug metabolism by providing key functions and metabolites to maximize the use of blood meals from these strict hematophagous bugs. However, the role of the microbiota in triatomine development remains poorly understood, in particular for *T. sanguisuga*, for which deficiencies in endosymbionts have been proposed to explain the poor survival of nymphs in captivity [23]. Our comparison of microbiota between laboratory-raised nymphs and field-caught adults provides initial support to this hypothesis and underlines metabolic differences that may explain the limited survival of

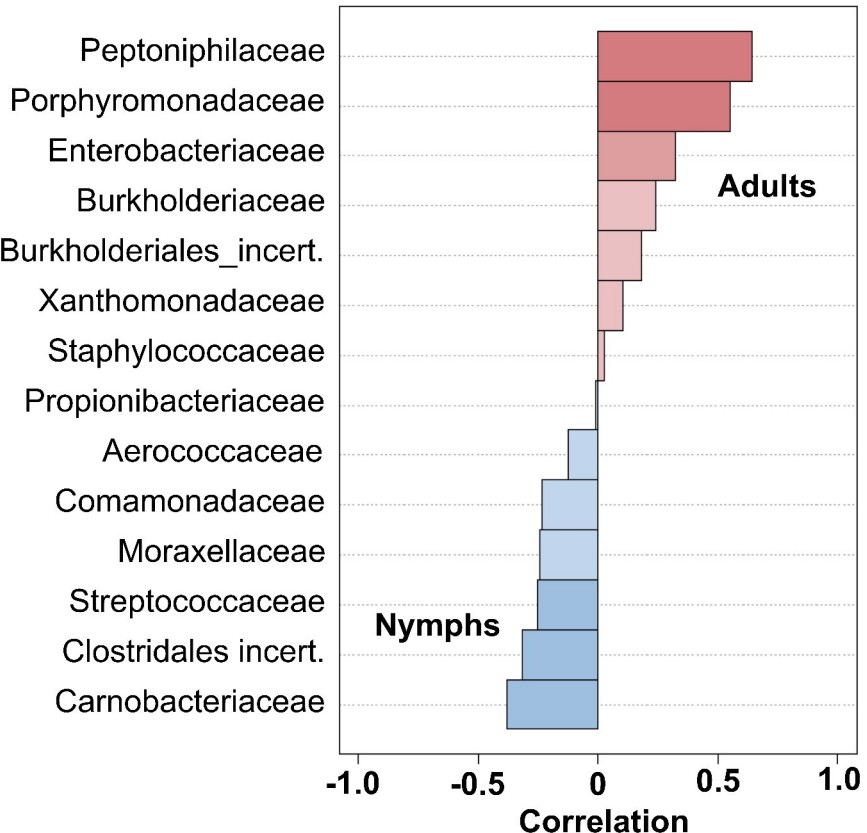

**Fig 3. Correlation analysis of bacterial taxa with developmental stage.** The association of bacterial taxa with developmental stage was assessed by sparCC correlation. Peptoniphilaceae, Porphyromonadaceae and to a lesser extent Enterobacteriaceae were significantly correlated with the field-caught adult microbiota, while Carnobacteriaceae, Clostridales incertae sedis, and Streptococcaceae were significantly correlated with laboratory-raised nymph microbiota ($P<0.05$).

nymphs in these conditions. Indeed, our data clearly show that there is no/limited vertical transmission of the field-caught adult microbiota to their laboratory-raised offspring, suggesting that nymphs nymphs need to acquire bacteria from their environment, and the ones they acquire under laboratory condition are insufficient for their development. Thus, the microbiota of *T. sanguisuga* seems to follow the general trend observed in other species of triatomines, with a more complex and diverse microbiota present in nymphs, while the adult microbiota is largely dominated by fewer taxa [17, 18, 20].

These data suggest a major reshuffle of the microbiota during development, with most taxa present in nymphs being replaced by a more limited set of taxa in adults, likely through coprophagy. However, at the functional level, the simpler adult microbiome of *T. sanguisuga* actually appears to provide a more complex metabolic network. The simplification of the microbiota in adults thus results in significant gains in metabolic functions. Indeed, the microbiota from laboratory-raised nymphs appears limited in its metabolic capabilities, with potential for competition among taxa, which may result in an unstable bacterial community of limited value to the bugs, leading to poor growth. On the other hand, the reduced complexity of the field-caught adult microbiota may allow for less competition, while at the same time ensuring a greater diversity of metabolic functions, which may better maximize the use of blood metabolites.

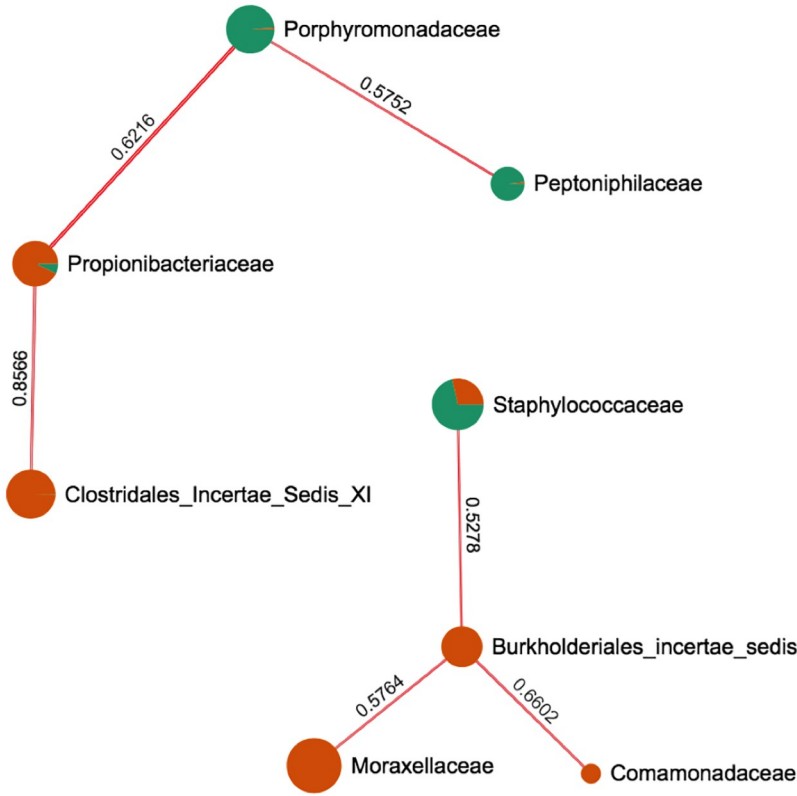

**Fig 4. Co-occurrence network of bacterial taxa in *T. sanguisuga*.** The co-occurrence of bacterial taxa in the microbiota of laboratory-raised nymphs (Orange circle nodes) and field-caught adults (Green circle nodes) was evaluated by SparCC correlations. The correlation coefficient is indicated on the edge linking two taxa. The size of the nodes is proportional to the relative abundance of the taxa in the microbiota.

However, it remains difficult to assess what key metabolic function(s) is/are provided by the filed-caught adult microbiota that is/are absent in the laboratory-raised nymph microbiota based on our predicted data. In *Rhodnius prolixus*, the supplementation of blood with vitamin B compounds can in part compensate for the absence of *Rhodococcus rhodnii* endosymbionts in laboratory-raised bugs [36]. It is however unclear what key metabolic function Peptoniphilaceae, Porphyromonadaceae and Enterobacteriaceae, which predominate in adult *T. sanguisuga*, may be providing to the bug. However, given the magnitude of the differences in bacterial taxa between nymphs and adults, it is likely that these bacteria are involved in multiple metabolic functions.

Triatomines also emerge as rather different from other Hemiptera, which as hemimetabolous insects are considered to have a rather stable microbiota during development [37]. Indeed, a dramatic reorganization of the gut microbiota during development has been proposed as a key feature of holometabolous insects, due to the major internal reconstructions during the pupal stage, which includes the replacement of the gut epithelium, while this process does not occur in hemimetabolous insects [37]. In that sense, the strict blood feeding of triatomine likely poses unique metabolic challenges, that may require more specific functions from their microbiota, hence more changes during bug development, which may be unique among Hemiptera and more generally in hemimetabolous insects.

There are however some limitations in our study. First, our limited sample size and the failure of bugs to molt past the third stage precluded an analysis through all laboratory-raised

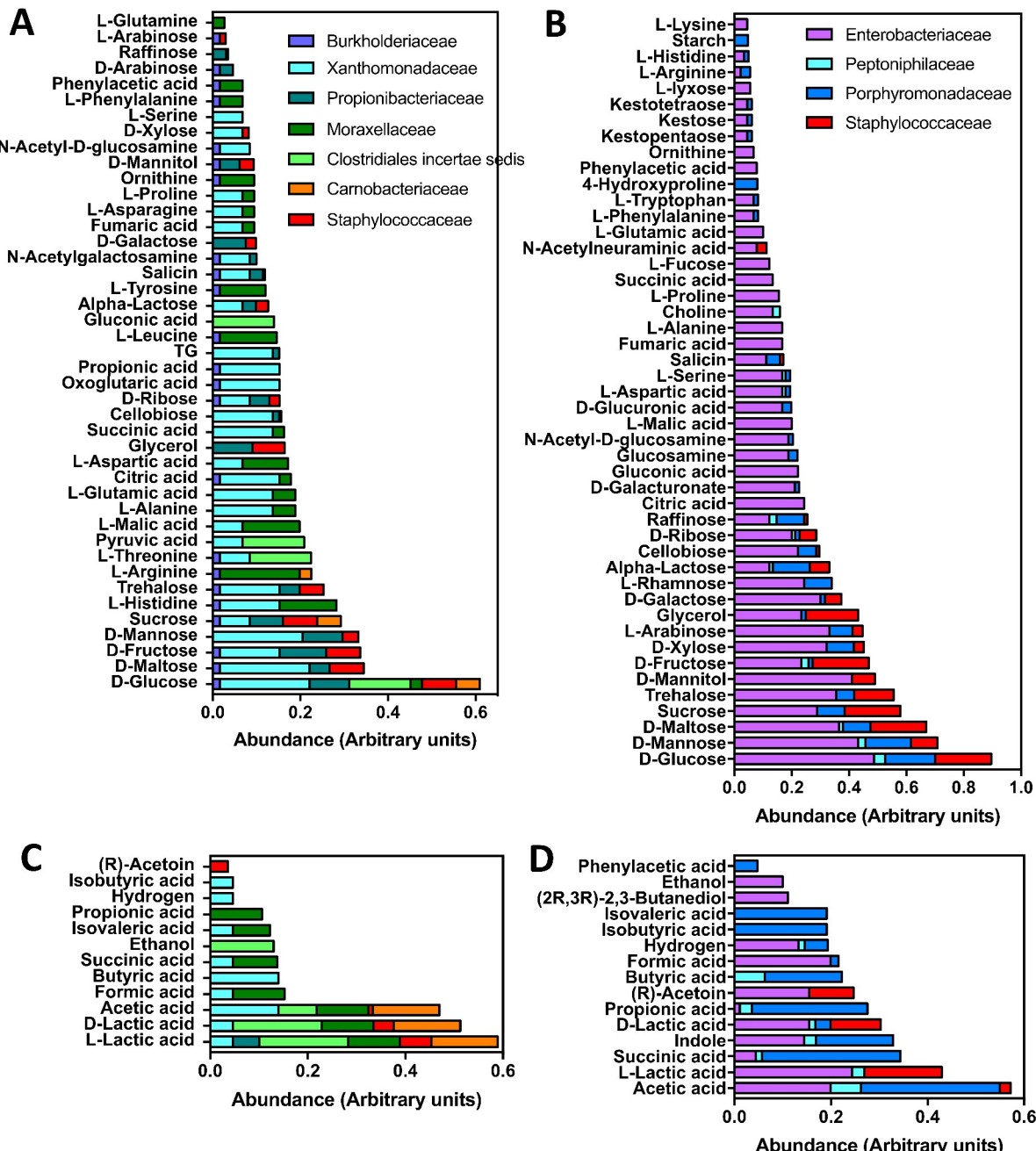

**Fig 5. Metabolic reconstruction of *T. sanguisuga* microbiota.** The relative abundance of carbon sources metabolized by nymph microbiota (**A**) is compared to those from the adult microbiota (**B**). Similarly, the relative amounts of fermentation products from the nymph microbiota (**C**) are compared with those from the adult microbiota (**D**). The relative contribution of the indicated bacterial taxa in the use/production of metabolites is color-coded as indicated.

nymphal stages and adults, as changes may be progressive as seen in *R. prolixus* [20], or more rapid. Also, only female adults were included, but previous work did not find significant differences between male and female microbiota in *T. sanguisuga* [4]. The inclusion of field-collected *T. sanguisuga* nymphs would also expand our analysis by providing further information on how these changes in microbiota occur in natural conditions. Finally, further refinement of

the microbiota at the species level would allow for more detailed metabolic predictions, as there is some heterogeneity in metabolic routes within bacterial families. Also, dissecting and analyzing different segments of the digestive tract of the bugs would allow further assessing potential differences in microbiota composition along the gut and the role of metabolic compartmentalization.

In conclusion, our study shows that *T. sanguisuga* nymphs with developmental failure have a diverse but metabolically limited microbiota, very different from that of field-collected adults. The adult microbiota, while including a lower bacterial diversity, provides a broader metabolic capacity that may be better suited to triatomine blood feeding diet. Further analysis, including of additional triatomine species, is warranted, as well as more detailed metabolic reconstructions of their microbiota for a better understanding of its role in triatomine biology and development.

## Supporting information

**S1 Fig. Rarefaction curves for individual samples.** Curves indicate adequate sequencing depth for samples included in the analysis.
(TIF)

**S2 Fig. Comparison of alfa and beta diversity of second and third stage nymphs.** For alpha diversity, there was no significant difference in Shannon (t = 0.45, *P* = 0.65) and Chao1 indices (t = 1.34, *P* = 0.21) between second (N2) and third stage nymphs (N3). Beta diversity was also not different between N2 and N3 (PERMANOVA, F = 1.9; P = 0.09).
(TIF)

## Author Contributions

**Conceptualization:** Evan Teal, Claudia Herrera, Eric Dumonteil.

**Data curation:** Evan Teal.

**Formal analysis:** Evan Teal, Claudia Herrera, Eric Dumonteil.

**Investigation:** Evan Teal, Claudia Herrera, Eric Dumonteil.

**Methodology:** Evan Teal, Eric Dumonteil.

**Supervision:** Claudia Herrera, Eric Dumonteil.

**Writing – original draft:** Evan Teal.

**Writing – review & editing:** Evan Teal, Claudia Herrera, Eric Dumonteil.

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
