## [Decision Letter · Decision Letter 0]

14 Dec 2022

PONE-D-22-29767Metabolomics of developmental changes in Triatoma sanguisuga gut microbiotaPLOS ONE

Dear Dr. Dumonteil,

Thank you for submitting your manuscript to PLOS ONE. After careful consideration, we feel that it has merit but does not fully meet PLOS ONE’s publication criteria as it currently stands. Therefore, we invite you to submit a revised version of the manuscript that addresses the points raised during the review process.

We look forward to receiving your revised manuscript.

Kind regards,

Yara M. Traub-Csekö

Academic Editor

PLOS ONE

Journal Requirements:

Reviewers' comments:

Reviewer's Responses to Questions

**Comments to the Author**

1. Is the manuscript technically sound, and do the data support the conclusions?

Reviewer #1: Partly

Reviewer #2: Partly

2. Has the statistical analysis been performed appropriately and rigorously? 

Reviewer #1: Yes

Reviewer #2: Yes

3. Have the authors made all data underlying the findings in their manuscript fully available?

Reviewer #1: Yes

Reviewer #2: Yes

4. Is the manuscript presented in an intelligible fashion and written in standard English?

Reviewer #1: Yes

Reviewer #2: Yes

5. Review Comments to the Author

Reviewer #1: Dear colleagues,

Identifying microbial diversity in Triatoma sanguisuga is interesting and adds important information to the adaptation scenario between wild-caught and laboratory-reared insects. The manuscript is well-written and can contribute to several other entomology research laboratories.

Nevertheless, a few aspects need to be clarified. Therefore, I hope my comments will help to improve the clarity of some aspects of the manuscript.

Major comments

The authors used two insect groups, one composed of field-caught adults and another group composed of their offspring reared under laboratory conditions, for analyzing the microbial diversity in the insect gut. I understand that rearing adults in the laboratory was not possible; therefore, nymphs were used in the analyses. Throughout the text, authors refer to these groups as nymphs and adults, reflecting differences in microbial diversity between the developmental stages of T. sanguisuga. Nevertheless, to address the difference between insect stages, authors should have compared nymphs and adults reared under the same conditions.

I presume that under the current experimental design, the changes in microbial diversity between wild-caught adults and laboratory-reared nymphs were likely caused by the adaptation to the laboratory conditions. This aspect needs to be clarified throughout the manuscript, including the title.

The authors sequenced the bacterial 16S ribosomal gene to identify the microbial diversity and consequently predict the metabolome. Therefore, it is an indirect finding. It also needs to let clearer in the manuscript.

Minor comments

I suggest using abbreviations like “field-caught adults (FCA) and laboratory-reared nymphs (LRN)” to avoid misunderstanding the comparison between the two insect groups. The way it is written throughout the manuscript, referring to nymphs and adults, may be misleading.

Line 54: citation number 8 does not match the references list.

Lines 73 and 74; 77 and 78: The context in this paragraph is the comparison between nymphs and adults. This comparison is possible if both stages are reared under the same conditions, evidencing that the changes occur through the development of the insects. Were authors aiming to use the same context in their analyses?

Line 104: The term “endosymbionts” must be checked if it is adequate here and throughout the manuscript. It is common that gut bacteria are referred to as endosymbionts, but not all of them can be clearly proven. Some of them are commensal, and others are opportunistic. In addition, several endosymbionts are intracellular and maternally transmitted. They have a lower probability of being lost across developmental stages.

Lines 149 to 151: In the present study, authors identified bacteria at the family level and deduced the metabolic reconstruction based on the association to a set of bacteria species available in the AGORA database. How often does a given bacterium share the same metabolic reconstruction with others from the same family? It is important to address the level of confidence in this association.

Discretionary comments

Line 111: Full name of “ASL2” was missed when used for the first time in the text.

Line 215: Would it be “variety” instead of “varied”?

Reviewer #2: Observations:

Line -25: I believe that instead of "alterative approaches" the authors mean "alternative approaches". Please fix it.

Line 54: it is spelled zoootic, but the correct one is zoonotic. please fix this

Line 55: change “synantropic” by “synanthropic”.

Line 56: the word “ authchtonous” is misspelled, change to the correct form “ autochthonous”.

Line 96: In the sentence “The paratransgenesis of triatomine microbiota …”, the authors are mistaken in relation to the concept of paratransgenic organism. It is not the microbiota that undergoes the paratransgenic process, but the vector in question. The microbiota of this vector undergoes a transgenic process. Who is paratransgenic is the vector, the modified microbiota is transgenic. I suggest changing the sentence to something like: "...the paratransgenic of triatomine vectors..." or "...the introduction of transgenic organisms into the microbiota of triatomine vectors..."

Lane 98: The word “edosymbonts” is misspelled, change to correct form “endosymbionts”.

Line 117: There is a mistake here, the sum of the number of nymphs in each stage (8+6+4) is 18 and not 11. Please fix it.

Line 136: “Micriobiota” is the misspelled form of “microbiota”. Please fix it.

Line 143: The singular verb was does not appear to agree with the plural subject Associations. Consider changing the verb form to were for subject-verb agreement.

Line 52: “… extracted, and weighted..” It appears that you have an unnecessary comma in a compound predicate. Consider removing it.

Line 215: The word varied doesn’t seem to fit this context. Consider replacing it with Variety

Line 231: ”… the role to the microbiota …” It seems that preposition use may be incorrect here. Consider change to of.

Line 277: the word occurr is a misspelling of occur. Fix it.

Figures:

The figures 3 and 4 are with the subtitles changed. Fix it.

Materials and methods:

To obtain the DNA to be used in the identification of the microbiota bacteria of T. sanguisuga. The authors used a section of the abdomen of adult insects and the entire body of nymphs. This generates noise, as in the case of nymphs bacteria that do not belong to the intestinal microbiota may be being counted as if they were. Ideally, material from nymphs and adults should be obtained through dissection of the intestines of both.

6. PLOS authors have the option to publish the peer review history of their article (what does this mean?). If published, this will include your full peer review and any attached files.

Reviewer #1: No

Reviewer #2: **Yes: **Antonio J Tempone

---

## [Author Response · Author response to Decision Letter 0]

28 Dec 2022

1. Is the manuscript technically sound, and do the data support the conclusions?

Reviewer #1: Partly

Reviewer #2: Partly

ANSWER: We thank the reviewers for their comments and have addressed these in detail below.

2. Has the statistical analysis been performed appropriately and rigorously? 

Reviewer #1: Yes

Reviewer #2: Yes

ANSWER: We thank the reviewers for their comments.

3. Have the authors made all data underlying the findings in their manuscript fully available?

Reviewer #1: Yes

Reviewer #2: Yes

ANSWER: We thank the reviewers for their comments.

4. Is the manuscript presented in an intelligible fashion and written in standard English?

Reviewer #1: Yes

Reviewer #2: Yes

ANSWER: We thank the reviewers for their comments.

5. Review Comments to the Author

Reviewer #1: Dear colleagues,

Identifying microbial diversity in Triatoma sanguisuga is interesting and adds important information to the adaptation scenario between wild-caught and laboratory-reared insects. The manuscript is well-written and can contribute to several other entomology research laboratories.

Nevertheless, a few aspects need to be clarified. Therefore, I hope my comments will help to improve the clarity of some aspects of the manuscript.

ANSWER: We thank the reviewer for their appreciation of our study and detail below how we have addressed the specific points raised.

Major comments

The authors used two insect groups, one composed of field-caught adults and another group composed of their offspring reared under laboratory conditions, for analyzing the microbial diversity in the insect gut. I understand that rearing adults in the laboratory was not possible; therefore, nymphs were used in the analyses. Throughout the text, authors refer to these groups as nymphs and adults, reflecting differences in microbial diversity between the developmental stages of T. sanguisuga. Nevertheless, to address the difference between insect stages, authors should have compared nymphs and adults reared under the same conditions.

I presume that under the current experimental design, the changes in microbial diversity between wild-caught adults and laboratory-reared nymphs were likely caused by the adaptation to the laboratory conditions. This aspect needs to be clarified throughout the manuscript, including the title.

ANSWER: We agree with the reviewer that we compare the microbiota of field-caught adults with that of their offspring reared under laboratory conditions, and it is indeed likely that laboratory conditions play a role in defining microbial composition in these bugs. Nonetheless, we clearly stated that our hypothesis was that “laboratory-raised nymphs lack key bacteria compared to field-collected adults”, which may help explain their failure to develop adequately in laboratory colonies. This comparison shows that there is no/limited vertical transmission of the field-caught adult microbiota to their lab-raised offspring, as we found that young (lab-raised) nymphs have a different microbiota from their (field-caught) parents. These data clearly indicate that nymphs need to acquire bacteria from their environment, and the ones they acquire under laboratory condition are insufficient for their development. These considerations have been added to the discussion for greater clarity (Page 10, lines 256-259). We also specifically mention as limitations of our study that “the failure of bugs to molt past the third stage precluded an analysis through all laboratory-raised nymphal stages and adults”, and that “the inclusion of field-collected T. sanguisuga nymphs would expand our analysis by providing further information on how these changes in microbiota occur in natural conditions.” (Page 12, lines 297-299, lines 301-303).

The authors sequenced the bacterial 16S ribosomal gene to identify the microbial diversity and consequently predict the metabolome. Therefore, it is an indirect finding. It also needs to let clearer in the manuscript.

ANSWER: We agree with the reviewer that the analysis of the metabolome is predicted from the microbiota composition, this is now better stressed in multiple sections of the manuscript for greater clarity (Abstract line 36; Page 5, Line 111; Page 7, line 171; Page 9, line 226; page 11, line 278; and page 12, line 304). 

Minor comments

I suggest using abbreviations like “field-caught adults (FCA) and laboratory-reared nymphs (LRN)” to avoid misunderstanding the comparison between the two insect groups. The way it is written throughout the manuscript, referring to nymphs and adults, may be misleading.

ANSWER: We agree with the reviewer that referring to field-caught adults and laboratory-reared nymphs is much more accurate than just adults and nymphs, and the manuscript has been edited accordingly for greater clarity. Note that we nonetheless prefer not to use the non-standard abbreviation proposed, to ensure that the manuscript remains easy to read for a general readership.

Line 54: citation number 8 does not match the references list.

ANSWER: The reference has been corrected.

Lines 73 and 74; 77 and 78: The context in this paragraph is the comparison between nymphs and adults. This comparison is possible if both stages are reared under the same conditions, evidencing that the changes occur through the development of the insects. Were authors aiming to use the same context in their analyses?

ANSWER: This paragraph summarizes the main findings from studies based on field-collected nymphs and adults from several species. This is now specified for greater clarity (Page 4, Line 80). These studies do suggest changes in microbiota during development from nymphs to adults, although rearing conditions were uncontrolled as these are field-caught bugs.

Line 104: The term “endosymbionts” must be checked if it is adequate here and throughout the manuscript. It is common that gut bacteria are referred to as endosymbionts, but not all of them can be clearly proven. Some of them are commensal, and others are opportunistic. In addition, several endosymbionts are intracellular and maternally transmitted. They have a lower probability of being lost across developmental stages.

ANSWER: We agree with the reviewer and have reworded this sentence to mention “bacteria” instead of “endosymbiont”.

Lines 149 to 151: In the present study, authors identified bacteria at the family level and deduced the metabolic reconstruction based on the association to a set of bacteria species available in the AGORA database. How often does a given bacterium share the same metabolic reconstruction with others from the same family? It is important to address the level of confidence in this association.

ANSWER: There is indeed some heterogeneity in metabolic routes within bacterial families, particularly with families including multiple genera and species, which we took into account by weighting metabolites based of the proportion of species within each family able to use/produce each metabolite. We now mention that “further refinement of the microbiota at the species level would allow for more detailed metabolic predictions” (Page 12, lines 303-305).

Discretionary comments

Line 111: Full name of “ASL2” was missed when used for the first time in the text.

ANSWER: This abbreviation has been removed for clarity.

Line 215: Would it be “variety” instead of “varied”?

ANSWER: Yes, wording has been changed as suggested.

Reviewer #2: Observations:

Line -25: I believe that instead of "alterative approaches" the authors mean "alternative approaches". Please fix it.

ANSWER: This is correct, spelling has been corrected.

Line 54: it is spelled zoootic, but the correct one is zoonotic. please fix this

ANSWER: Spelling has been corrected.

Line 55: change “synantropic” by “synanthropic”.

ANSWER: Spelling has been corrected.

Line 56: the word “ authchtonous” is misspelled, change to the correct form “ autochthonous”.

ANSWER: Spelling has been corrected.

Line 96: In the sentence “The paratransgenesis of triatomine microbiota …”, the authors are mistaken in relation to the concept of paratransgenic organism. It is not the microbiota that undergoes the paratransgenic process, but the vector in question. The microbiota of this vector undergoes a transgenic process. Who is paratransgenic is the vector, the modified microbiota is transgenic. I suggest changing the sentence to something like: "...the paratransgenic of triatomine vectors..." or "...the introduction of transgenic organisms into the microbiota of triatomine vectors..."

ANSWER: We agree with the reviewer and have edited this sentence accordingly.

Lane 98: The word “edosymbonts” is misspelled, change to correct form “endosymbionts”.

ANSWER: Spelling has been corrected.

Line 117: There is a mistake here, the sum of the number of nymphs in each stage (8+6+4) is 18 and not 11. Please fix it.

ANSWER: The number has been corrected.

Line 136: “Micriobiota” is the misspelled form of “microbiota”. Please fix it.

ANSWER: Spelling has been corrected.

Line 143: The singular verb was does not appear to agree with the plural subject Associations. Consider changing the verb form to were for subject-verb agreement.

ANSWER: Spelling has been corrected.

Line 52: “… extracted, and weighted..” It appears that you have an unnecessary comma in a compound predicate. Consider removing it.

ANSWER: The coma has been removed.

Line 215: The word varied doesn’t seem to fit this context. Consider replacing it with Variety

ANSWER: Spelling has been corrected.

Line 231: ”… the role to the microbiota …” It seems that preposition use may be incorrect here. Consider change to of.

ANSWER: Wording has been corrected as suggested.

Line 277: the word occurr is a misspelling of occur. Fix it.

ANSWER: Spelling has been corrected.

Figures:

The figures 3 and 4 are with the subtitles changed. Fix it.

ANSWER: The figure legends are correct, but figure 3 and 4 were swapped, and they have now been renumbered.

Materials and methods:

To obtain the DNA to be used in the identification of the microbiota bacteria of T. sanguisuga. The authors used a section of the abdomen of adult insects and the entire body of nymphs. This generates noise, as in the case of nymphs bacteria that do not belong to the intestinal microbiota may be being counted as if they were. Ideally, material from nymphs and adults should be obtained through dissection of the intestines of both.

ANSWER: We agree with the reviewer that the inclusion of non-gut bacteria may have occurred, but this bias would be similar for both adult and nymphs as both samples included some other tissues and external cuticle. Such bias is also likely small as most bacterial families identified were anaerobic bacteria unlikely to derive from non-gut sources. We nonetheless added to the discussion that “dissecting and analyzing different segments of the digestive tract of the bugs would allow further assessing potential differences in microbiota composition along the gut and the role of metabolic compartmentalization” (Page 12, lines 305-308).

---

## [Decision Letter · Decision Letter 1]

11 Jan 2023

Metabolomics of developmental changes in Triatoma sanguisuga gut microbiota

PONE-D-22-29767R1

Dear Dr. Dumonteil,

We’re pleased to inform you that your manuscript has been judged scientifically suitable for publication and will be formally accepted for publication once it meets all outstanding technical requirements.

Kind regards,

Yara M. Traub-Csekö

Academic Editor

PLOS ONE

Additional Editor Comments (optional):

Reviewers' comments:

Reviewer's Responses to Questions

**Comments to the Author**

1. If the authors have adequately addressed your comments raised in a previous round of review and you feel that this manuscript is now acceptable for publication, you may indicate that here to bypass the “Comments to the Author” section, enter your conflict of interest statement in the “Confidential to Editor” section, and submit your "Accept" recommendation.

Reviewer #1: All comments have been addressed

Reviewer #2: All comments have been addressed

2. Is the manuscript technically sound, and do the data support the conclusions?

Reviewer #1: Yes

Reviewer #2: Yes

3. Has the statistical analysis been performed appropriately and rigorously? 

Reviewer #1: Yes

Reviewer #2: Yes

4. Have the authors made all data underlying the findings in their manuscript fully available?

Reviewer #1: Yes

Reviewer #2: Yes

5. Is the manuscript presented in an intelligible fashion and written in standard English?

Reviewer #1: Yes

Reviewer #2: Yes

6. Review Comments to the Author

Reviewer #1: Dear colleagues,

The page and line numbers cited in the authors' answers were not matching with the new submitted version. Nevertheless, the corresponding changes could be found.

All questions and comments were addressed.

Reviewer #2: (No Response)

7. PLOS authors have the option to publish the peer review history of their article (what does this mean?). If published, this will include your full peer review and any attached files.

Reviewer #1: No

Reviewer #2: **Yes: **Antonio J Tempone

---

## [Editor Report · Acceptance letter]

15 Feb 2023

PONE-D-22-29767R1 

Metabolomics of developmental changes in *Triatoma sanguisuga* gut microbiota 

Dear Dr. Dumonteil:

I'm pleased to inform you that your manuscript has been deemed suitable for publication in PLOS ONE. Congratulations! Your manuscript is now with our production department. 

Kind regards, 

on behalf of

Dr. Yara M. Traub-Csekö 

Academic Editor

PLOS ONE